# Metabolic and chromosomal changes in a *Bacillus subtilis* *whiA* mutant

Laura C. Bohorquez,[1] Joana de Sousa,[2] Transito Garcia-Garcia,[3] Gaurav Dugar,[1] Biwen Wang,[1] Martijs J. Jonker,[4] Marie-Françoise Noirot-Gros,[3] Michael Lalk,[2] Leendert W. Hamoen[1]

**ABSTRACT** The conserved protein WhiA is present in most Gram-positive bacteria and plays a role in cell division. WhiA contains a DNA-binding motif and is a transcription regulator of the key cell division gene *ftsZ* in actinomycetes. In *Bacillus subtilis*, the absence of WhiA influences both cell division and chromosome segregation; however, the protein does not regulate any gene involved in these processes. In this study, we addressed three alternative mechanisms by which WhiA might exert its activity in *B. subtilis* and examined whether WhiA influences either (i) central carbon metabolism, (ii) fatty acid composition of the cell membrane, or (iii) chromosome organization. Mutations in glycolytic enzymes have been shown to influence both cell division and DNA replication. To measure the effect of WhiA on carbon metabolism, we tested different carbon sources and measured exometabolome fluxes. This revealed that the absence of WhiA does not affect glycolysis but does influence the pool of branched-chain fatty acid precursors. Due to the effect of WhiA on chromosome segregation, we examine chromosome organization in a Δ*whiA* mutant using chromosome conformation capture (Hi-C) analysis. This revealed a local reduction in short-range chromosome interactions. Together, these findings provide new avenues for future research into how this protein works in the non-actinomycete firmicutes.

**IMPORTANCE** WhiA is a conserved DNA-binding protein that influences cell division in many Gram-positive bacteria and, in *B. subtilis,* also chromosome segregation. How WhiA works in *Bacillus subtilis* is unknown. Here, we tested three hypothetical mechanisms using metabolomics, fatty acid analysis, and chromosome confirmation capture experiments. This revealed that WhiA does not influence cell division and chromosome segregation by modulating either central carbon metabolism or fatty acid composition. However, the inactivation of WhiA reduces short-range chromosome interactions. These findings provide new avenues to study the molecular mechanism of WhiA in the future.

**KEYWORDS** WhiA, *Bacillus subtilis*, cell division, branched-chain fatty acids, chromosome compaction

Address correspondence to Leendert W. Hamoen, l.w.hamoen@uva.nl.

The authors declare no conflict of interest.

See the funding table on p. 16.

WhiA is a conserved DNA-binding protein present in most Gram-positive bacteria, including the simple cell wall-lacking Mycoplasmas. The crystal structure of *Thermotoga maritima* WhiA shows a bipartite conformation in which a degenerate N-terminal LAGLIDADG homing endonuclease domain is tethered to a C-terminal helix-turn-helix DNA-binding domain. However, none of the characterized WhiA proteins show any nuclease activity (1). In the actinomycetes *Streptomyces coelicolor*, *Streptomyces venezuelae,* and *Corynebacterium glutamicum,* it has been shown that WhiA functions as a transcriptional activator of several genes, among which the key cell division gene *ftsZ* (2–5). Inactivation of WhiA in *S. coelicolor* prevents the induction of FtsZ, thereby blocking the synthesis of sporulation septa (6, 7). Recently, it was shown that

the N-terminal homing endonuclease-like domain of WhiA interacts with the protein WhiB, forming a complex that stimulates transcriptional activation (8). In *Bacillus subtilis,* inactivation of WhiA reduces the growth rate in rich medium and affects the expression of various genes but not that of *ftsZ* or other cell division-related genes (9). WhiB is unique to the actinomycetes, and *B. subtilis* lacks a WhiB homolog. There is also no correlation between WhiA-binding sites on the chromosome and genes that show a different expression when *whiA* is inactivated, suggesting that WhiA does not function as a classic transcription factor in this organism (9). Nevertheless, WhiA is important for cell division in *B. subtilis*, and the absence of WhiA is lethal when cell division proteins are inactivated, which regulates the formation of the Z-ring, including the MinCD proteins that prevent aberrant division leading to minicell formation, or Noc, responsible for the nucleoid occlusion of FtsZ polymerization, or the FtsZ polymer crosslinker ZapA (9). Later, it was found that WhiA is also important for proper chromosome segregation in *B. subtilis*, and a ΔwhiA mutant displays increased nucleoid spacing and is synthetically lethal when either the DNA replication regulator ParB or the DNA replication inhibitor YabA is absent (10). It is still unclear how WhiA operates in *B. subtilis* and other Gram-positive non-actinomycetes. Many different processes influence cell division and chromosome replication, including carbon metabolism, the lipid composition of the cell membrane, and chromosome organization. In this study, we have explored whether the activity of WhiA is related to one of these processes.

## RESULTS AND DISCUSSION

### Growth on different carbon sources

The earliest event in bacterial cell division is the polymerization of the tubulin-like protein FtsZ at midcell. Much of our understanding of bacterial cell division has been derived from heat-sensitive FtsZ mutants. Interestingly, inactivation of the glycolytic enzyme pyruvate kinase can rescue some of these FtsZ mutants (11). Deletion of phosphoglucomutase, which catalyzes the interconversion of glucose-1-phosphate and glucose-6-phosphate, also suppresses certain heat-sensitive FtsZ mutants, both in *B. subtilis* and *Escherichia coli* (12). Mutations in different glycolytic enzymes, among which pyruvate kinase, can also suppress temperature-sensitive mutants in DNA initiation and replication proteins in different bacteria (13). How these glycolytic mutants rescue certain cell division and chromosome replication mutants is not yet clear. Since the inactivation of WhiA affects both cell division and DNA segregation, it is tempting to speculate that WhiA influences cell division and chromosome replication by playing a role in central carbon metabolism. This hypothesis is further supported by the fact that in many bacteria *whiA* is located adjacent to the gene *glmR* and *crh* (14). GlmR has been shown to be essential for growth and normal cell shape under gluconeogenic growth conditions (15). This protein regulates carbon partitioning between central carbon metabolism and peptidoglycan biosynthesis (16). *crh* codes for the carbon-flux regulator HPr that regulates the methylglyoxal synthase MgsA, which provides a bypass of glycolysis when phosphorylated glycolytic intermediates accumulate in the cell (17). Interestingly, elevated methylglyoxal levels affect cell division, possibly by reducing *ftsZ* expression (18). In *B. subtilis*, the *whiA* gene forms an operon together with *glmR* and *crh*, located upstream and downstream of *whiA*, respectively, and this conserved organization might be related to a shared functional activity, which could mean that the cell division and possible DNA segregation phenotype of a *whiA* mutant is related to a change in carbon metabolism.

Inactivation of *glmR* blocks growth on citrate and results in very poor growth on either fumarate or malate as sole carbon sources (19). To examine whether the absence of WhiA also affects growth on these carbon sources, we grew a ΔwhiA mutant in Spizizen minimal salt medium (SMM) using either malate, fumarate, or citrate as a carbon source. To prevent any downstream effects, a marker-less ΔwhiA mutant was used, containing a stop codon at the beginning of the gene [strain KS696 (9)]. As shown in Fig. 1, the ΔwhiA mutant was able to grow in these different media with growth rates

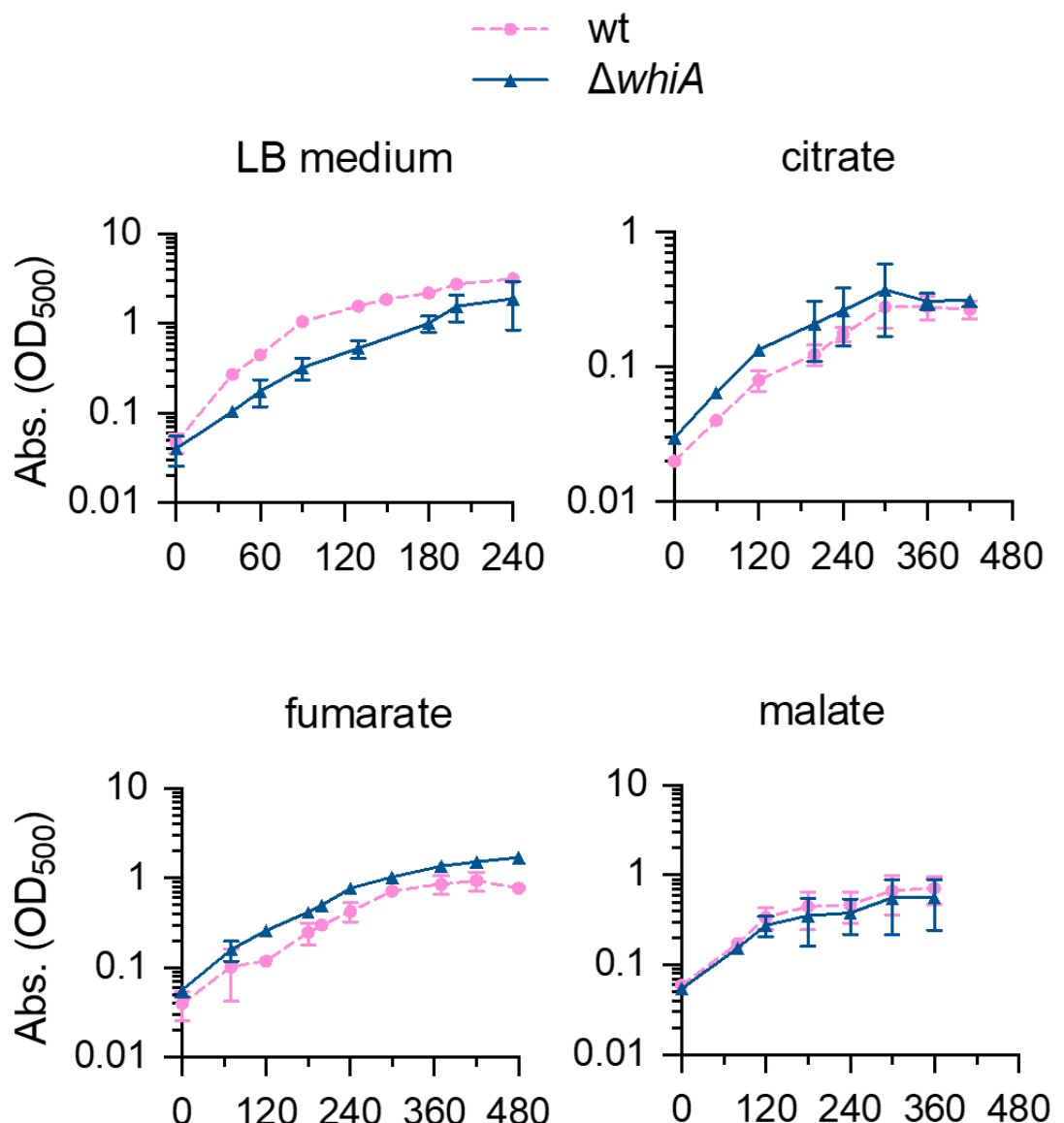

**FIG 1**  Growth of a Δ*whiA* strain on different carbon sources. Growth of the wild-type strain (strain 168) and the *whiA* marker-less mutant (strain KS696) in rich Luria-Bertani (LB) medium and in (SMM supplemented with 22 mM of either citrate, fumarate, or malate. Data are shown as mean values and standard deviation of triplicate samples.

similar to that of the wild-type strain, indicating that WhiA and GlmR work in different pathways.

## Cell division and chromosome segregation phenotype in minimal medium

As shown in Fig. 1, and reported before, the Δ*whiA* mutant grows slower in rich Luria-Bertani (LB) medium (9). The reason that this reduction in growth rate was not observed in the minimal medium (Fig. 1) could be related to the lower doubling time in the minimal medium compared to LB (~53 min versus ~21 min), which can mitigate chromosome segregation and cell division defects (20, 21). Previously, we have shown that removing WhiA in a strain lacking the FtsZ crosslinker ZapA interferes with the formation of Z-rings and blocks cell division (9). To examine whether this synthetic phenotype could be abrogated by growing cells in a minimal medium, we first tried to obtain a *whiA zapA* double deletion mutant by using minimal medium plates to select transformants, but this did not yield viable colonies. Subsequently, we tested the

effect of WhiA depletion in a Δ*zapA* strain by placing *whiA* under the control of the isopropyl β- d-1-thiogalactopyranoside (IPTG)-inducible *Pspac* promoter. This strain was grown in a minimal medium without IPTG and although no clear effect on growth was observed over the measured period (Fig. 2A), as has been observed before in a depletion experiment in a rich medium (10), over time cells became elongated, indicative of a defect in cell division and occasionally generated aberrant nucleoids (Fig. 2A). Of note, these cell division defects are not caused by downregulation of the downstream located *crh* gene (9). Also, the inter-nucleoid spacing in a Δ*whiA* mutant grown in a minimal medium was still larger than the wild-type cells (Fig. 2B), thus slower growth in a minimal medium does not overcome the cell division and DNA segregation phenotypes of a Δ*whiA* mutant.

## Utilization of carbon sources during growth

As outlined above, changes in glycolysis can affect both cell division and chromosome replication. To examine whether inactivation of WhiA affects glycolysis, we measured the carbon consumption by means of exometabolomics, using proton nuclear magnetic resonance spectroscopy ([1]H-NMR) (22). This required a minimal chemically defined medium, for which M9 medium is often used. However, the M9 medium has been optimized for *E. coli* and not for *B. subtilis*, and the latter easily lyses in this medium in the stationary phase (22). Therefore, we composed an alternative chemically defined medium based on different minimal media used for *B. subtilis*, as listed in Table S1. In essence, the resulting medium, named Amber medium, uses a phosphate buffer, and ammonium salt and glutamate as nitrogen sources. The concentration of different carbon sources tested was 22 mM. To determine the exometabolome, 2 mL of culture was collected at regular time intervals and rapidly filtered for [1]H-NMR spectroscopic analysis. Identification of metabolites was based on NMR spectra alignment of pure standard compounds, and the quantification was done based on the integration, and comparison of the designated peaks to an internal standard peak (see the Materials and Methods section for details). The final results were based on three independent biological replicates, and the quality of the replicates was confirmed using a principal component analysis (PCA) (Fig. S1). First, we tested the consumption of different carbon sources in the presence of glucose to check for possible effects on catabolite repression. Cells were grown in Amber medium containing glucose alone, glucose and citrate, glucose and fumarate, and glucose and malate. Both the wildtype and the marker-less Δ*whiA* mutant grew fine in these media (Fig. S3). Malate was incorporated in this analysis since it is the second preferred carbon source of *B. subtilis*, and its utilization is not subjected to carbon catabolite repression in this organism (23). As shown in Fig. 3, citrate and fumarate utilization was initiated when most glucose was exhausted, confirming that fumarate and citrate were subjected to glucose-dependent catabolite repression in both strains. Malate was consumed faster than glucose as has been shown before (22). There was no apparent difference between the wild-type and Δ*whiA* strains, indicating that WhiA does not influence catabolite repression.

## Exometabolome analysis with different carbon sources

Aside from the supplied carbon sources (glucose, citrate, fumarate, and malate), we were able to detect 18 other metabolites in the medium. To facilitate the interpretation of the exometabolome data, the time-resolved extracellular metabolite concentrations were plotted onto the relevant pathways. As shown in Fig. 4 and 5, the levels of pyruvate, although differing substantially between growth conditions, were comparable for the wild-type and the Δ*whiA* strains, indicating that, at least under the conditions tested here, WhiA does not change the efficiency of glycolysis.

After approximately 3 h of growth, when glucose levels started to go down, Δ*whiA* mutant cells started to show a reduced level of the branched-chain fatty acid precursors, isovalerate, isobutyrate, and 2-methylbutyrate, and an increase in acetate and 2-oxoglutarate. We were not able to identify isoleucine, leucine, and oxaloacetate due to the

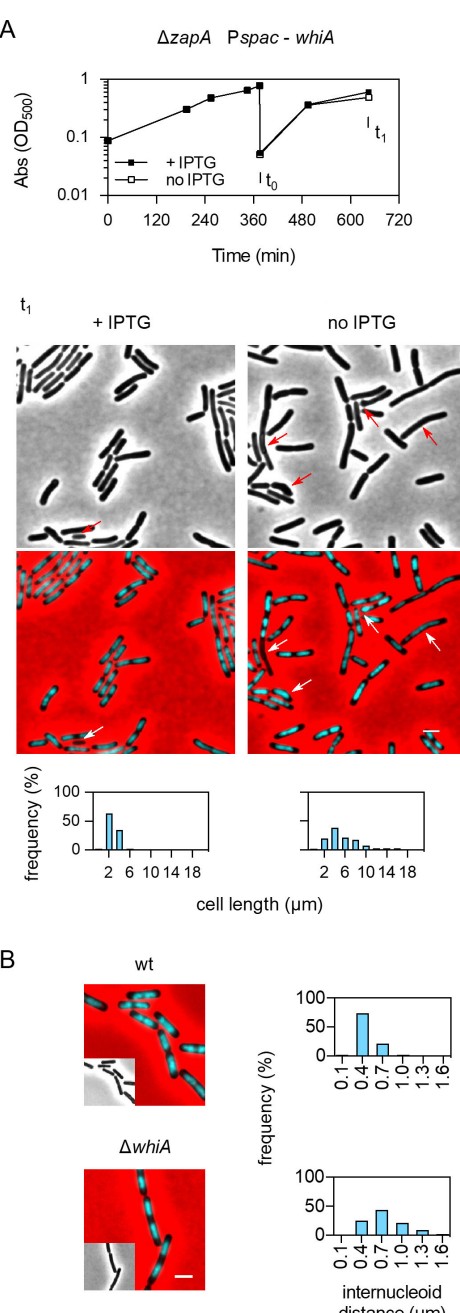

**FIG 2** Cell division and chromosome segregation defects of Δ*whiA* cells in minimal medium. (A) Growth curves of strain LB45 containing a Δ*zapA* mutation and an IPTG-inducible *whiA* allele. The strain was grown in minimal SMM in the presence of 0.1 mM IPTG to an $OD_{600}$ of 0.8. After that, the culture was diluted into fresh medium in the presence or absence of 0.1 mM IPTG (arrow at $t_0$). Middle panel shows fluorescence and phase contrast microscopy composite pictures of the cells after 270 min ($t_1$). Nucleoids were stained with 4′,6-diamidino-2-phenylindole (cyan), and the phase contrast images were pseudocolored red to enhance the contrast. Scale bar is 2 μm. Arrows indicate cells with aberrant nucleoids and a-nucleate cells. Lower panels show the cell length distribution (*n* = 307 and 385 for +IPTG and no IPTG, respectively). (B) Fluorescence and phase contrast microscopy composite picture of exponentially growing wild-type cells and Δ*whiA* cells (strain KS696) cultivated in minimal SMM at 37°C and stained with DAPI (cyan) to mark nucleoids. The phase contrast images are pseudocolored red to enhance the contrast. Scale bar is 2 μm. Right panels show the inter-nucleoid distances (*n* = 607 and 835 for wild-type and Δ*whiA* cells, respectively).

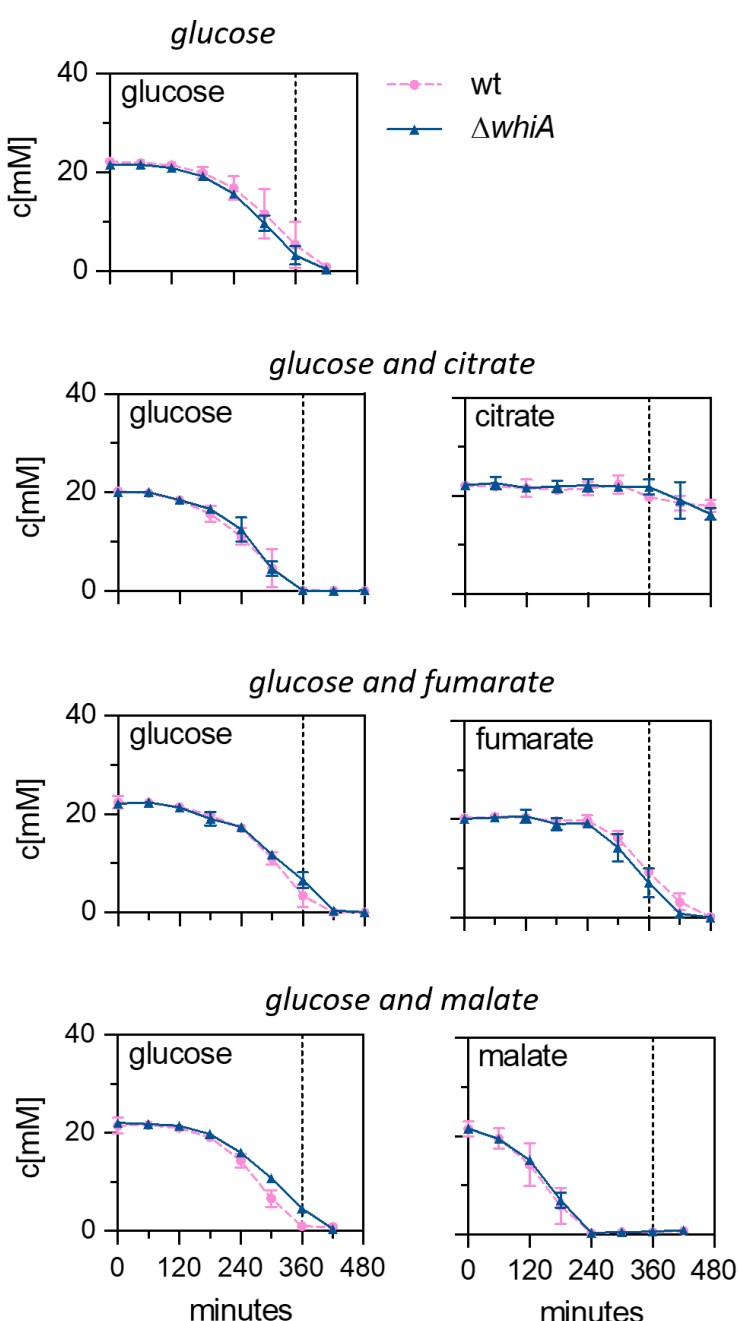

**FIG 3** Carbon utilization over time. Carbon source utilization (concentration in mM) of wild-type (strain 168) and *whiA* marker-less mutant cells (strain KS696) during growth in defined minimal Amber medium supplemented with either glucose, glucose and citrate, glucose and fumarate, or glucose and malate. Data are shown as mean values and standard deviation of triplicate samples. The dashed lines mark the time point when the glucose culture enters the stationary phase (360 min) (see Fig. S3).

detection limits of the method (22). Citrate and isocitrate were only measurable when the medium contained the trichloroacetic acid (TCA) intermediate citrate or fumarate (Fig. 4, lower panel, and Fig. 5, upper panel). The reason for the latter could be that the expression of citrate synthase and aconitase is induced when citrate is present in the medium or when fumarate becomes the sole carbon source after glucose levels have fallen (22, 24).

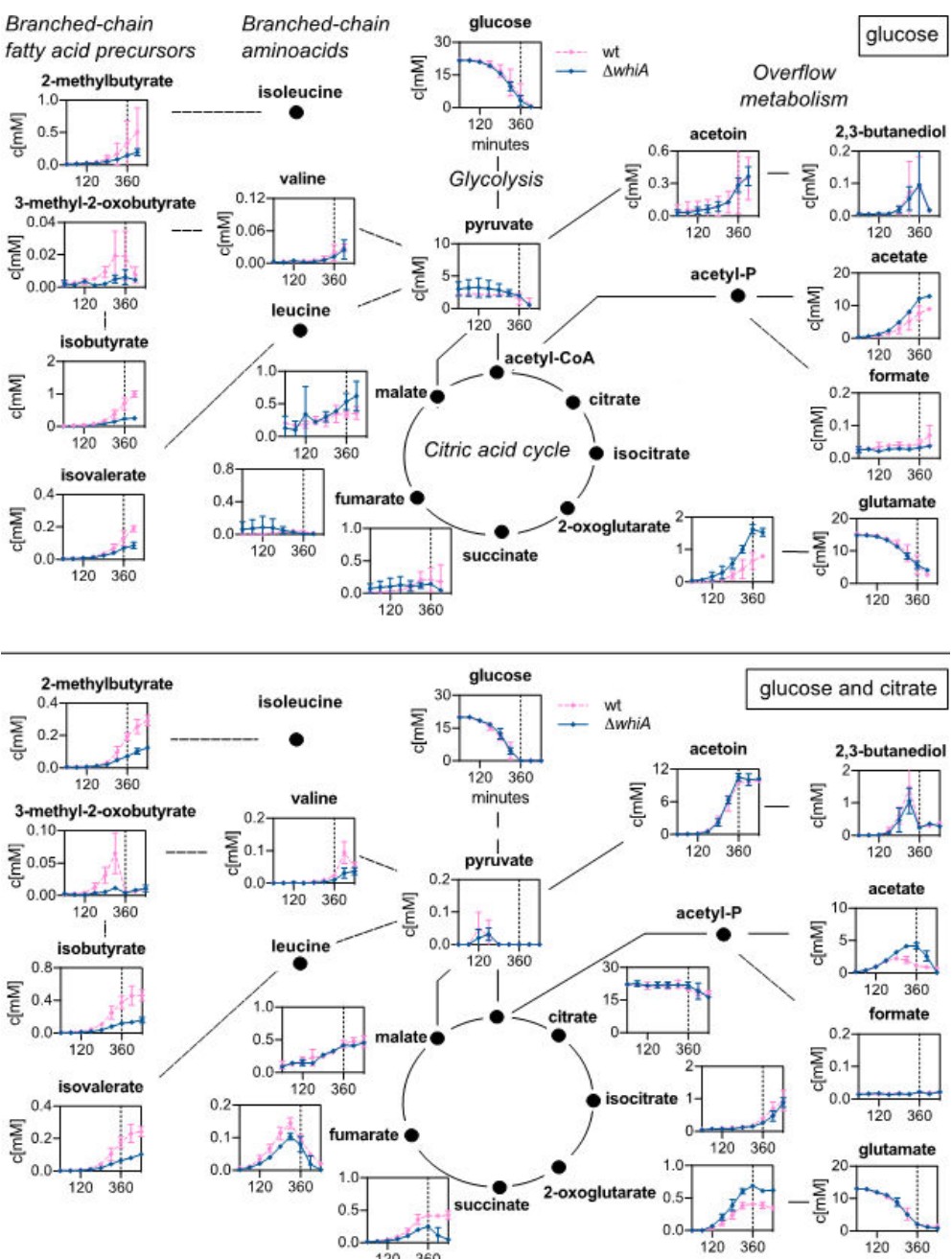

**FIG 4** Exometabolome of cells grown with either glucose or glucose and citrate. Time-resolved extracellular metabolite concentrations (in mM) of wild-type and Δ*whiA* marker-less mutant cells (strain KS696) grown in chemically defined minimal Amber medium with either glucose alone (upper panel) or glucose and citrate (lower panel) as carbon sources. Dashed lines indicate entry into the stationary phase (360 min). The compounds are arranged according to the main metabolic pathways: glycolysis, TCA cycle, overflow metabolites, branched-chain amino acids, and branched-chain fatty acids precursors. Data are shown as mean values and standard deviation of triplicate samples.

## Transcriptome analysis with glucose and malate as carbon sources

To examine whether the changes in metabolism were related to changes in gene expression, we compared the transcriptomes of wild-type and Δ*whiA* mutant cells grown in Amber medium supplemented with glucose and malate as carbon sources, when the differences in branched-chain fatty acid precursors became noticeable, i.e., around an OD$_{500}$ of 0.5. The medium with glucose and malate was chosen since this combination

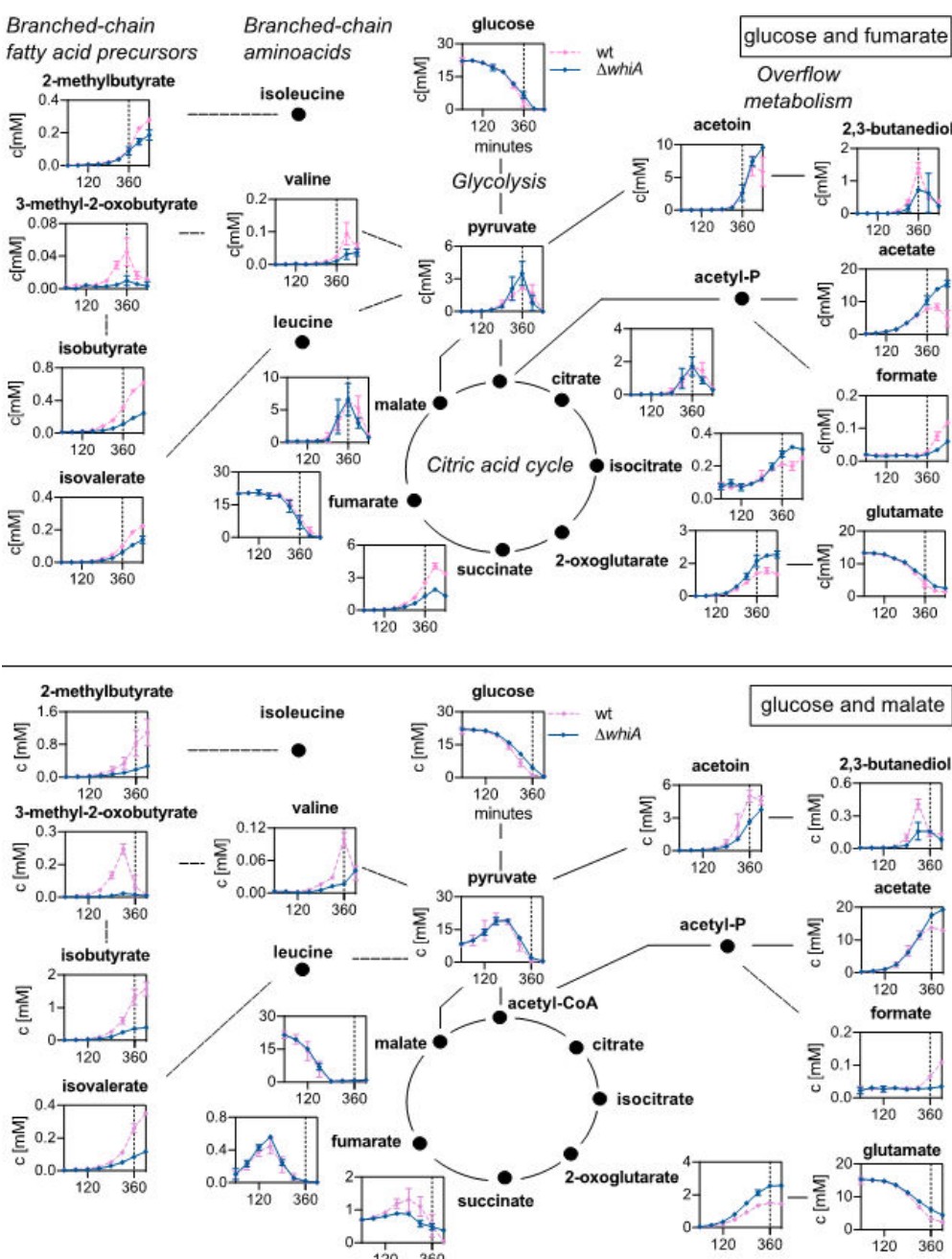

**FIG 5** Exometabolome in cells grown with either glucose and fumarate or glucose and malate. Time-resolved extracellular metabolite concentrations (in mM) of wild-type and Δ*whiA* marker-less mutant cells (strain KS696) grown in chemically defined minimal Amber medium with either glucose and fumarate (upper panel) or glucose and malate (lower panel) as carbon sources. Dashed lines indicate entry into the stationary phase (360 min). The compounds are arranged according to the main metabolic pathways: glycolysis, TCA cycle, overflow metabolites, branched-chain amino acids, and branched-chain fatty acid precursors. Data are shown as mean values and standard deviation of triplicate samples.

showed the clearest difference in branched-chain fatty acid precursors. The experiment was repeated to provide a biological replicate. The volcano plot in Fig. 6 depicts the distribution of expression differences against adjusted *P*-values. A total of 57 genes were upregulated and 40 downregulated more than threefold when an adjusted *P*-value < 0.05 was used as a threshold value (Table 1, data for all genes are listed in Table S6). The most highly upregulated genes, *ydcF*, *ydcG*, and *pamR*, form an operon. PamR is a

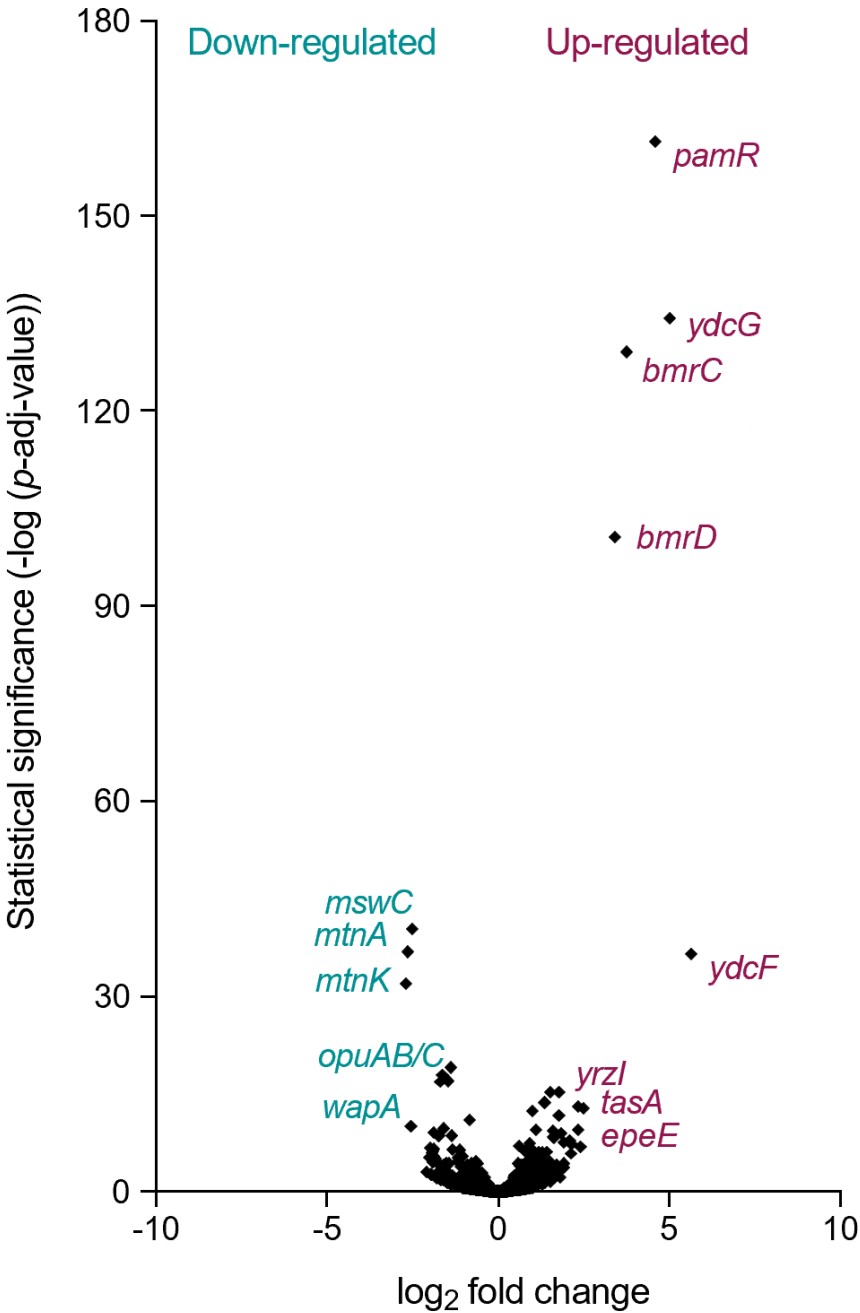

**FIG 6** Volcano plot of transcriptome data. Volcano plot depicting the transcriptome data as a relation between adjusted *P*-values and log$_2$ fold expression change. Wild-type and *whiA* marker-less mutant cells (strain KS696) were grown in a defined minimal Amber medium with glucose and malate and sampled during exponential growth. Main downregulated and upregulated genes in the *whiA* mutant are shown in cyan and magenta, respectively.

transcription factor that affects the expression of prophages and certain metabolic genes (25). Another strongly upregulated operon is the *bmrB* operon, coding for a multidrug ABC transporter (26). This transporter is also involved in the activation of KinA, one of the key regulators of sporulation (27). Of note, a Δ*whiA* mutant displays only a very mild defect in sporulation (9). The upregulated *tapA* operon is required for the synthesis of the major extracellular matrix (28). Other upregulated genes were the *epeX* operon encoding proteins controlling the activity of the LiaRS cell envelope stress-response system (29), the *fatR* operon involved in lipid degradation (30), and the *sunA* and *nupN* operons

**TABLE 1** Transcriptome comparison of wild-type and Δ*whiA* cells[a]

| Gene | FC (Δ*whiA*/wt) | Function |
|---|---|---|
| Upregulated | | |
| ydcF*-G*-pamR*[b] | 24–50 | Unknown |
| bmrB-C*-D* | 4–14 | Multidrug ABC transporter |
| tapA*-sipW*-tasA* | 4–6 | Major component of biofilm matrix |
| yxbB-A-yxnB-asnH-yxaM | 3–5 | Biosynthesis of asparagine and unknown |
| yxbC-D | 3 | Unknown (upstream of yxbB operon) |
| yrzl | 5 | Unknown |
| epeX-E-P-A-B | 2–5 | Control of LiaRS cell envelope stress system |
| ybdZ | 3 | Unknown |
| yfmG | 3 | Unknown |
| fatR-yrhJ | 3 | Fatty acid metabolism |
| dhbA*-C*-E*-B*-F* | 3 | Biosynthesis of the siderophore bacillibactin |
| besA* | 2 | Iron acquisition, ferri-bacillibactin esterase |
| yobB | 4 | Unknown |
| sunA-T-bdbA*-sunA-bdbB | 3–4 | Sublancin lantibiotic production and thiol-disulfide oxidoreductase |
| yitP-O-M | 3 | Biofilm toxin and unknown |
| nupN-O-P | 3 | Uptake of guanosine |
| yoaW | 3 | Secreted protein with unknown function |
| ybdN | 3 | Unknown |
| yyzl | 3 | Unknown |
| skfA-B-C-E-F-G-H | 2–4 | Spore killing factor |
| Downregulated | | |
| mtnU*-A*-K* | -(3–6) | Methionine salvage |
| wapA*-I*-yxzC*-G*-J*-I*-yxiG*-H*-I*-J*-K*-M* | -6 | Cell wall-associated WapA protein toxin & unknown |
| yonN-J-B-yomW-U-Z | -(3–4) | Parts of SP-beta prophage genome |
| fadN-A-E | -4 | Fatty acid degradation |
| bsdB-C-yclD | -(3–4) | Resistance to salicylic acid |
| Tdh-kbl | -(3–4) | Threonine utilization |
| frlB-O-N-M | -(3–4) | Fructose metabolism |
| yezD | -3 | Unknown |
| proH-J | -3 | Osmoadaptive *de novo* production of proline |
| oxdC | -3 | Oxalate decarboxylase |
| licH | -3 | 6-Phospho-beta-glucosidase, lichenan utilization |
| citZ | -3 | Citrate synthase, TCA cycle |

[a]Cells were grown in a defined minimal Amber medium with glucose and malate and harvested for RNA isolation during exponential growth (OD600 ~0.5). A marker-less ΔwhiA mutant was used (strain KS696). Genes with an adjusted *P*-value < 0.05 and fold change (FC) > 3 are listed. The fold change was calculated by Δ*whiA*/wt. Genes found in a previous ΔwhiA transcriptome analysis performed in LB rich medium are indicated by * (9) (see also Fig. S4). Genes located in the same operon are listed together in one row. In this case, the maximum and minimum FC values are indicated.
[b]Genes found in a previous whiA transcriptome analysis performed in LB rich medium are indicated by *

involved in the biosynthesis of an antimicrobial peptide and the uptake of guanosine, respectively (31–33).

Several genes involved in amino acid biosynthesis were downregulated, including the *mtnA* operon involved in methionine salvage (34), the *tdh* operon involved in threonine utilization (35, 36), and *proHJ* necessary for the production of proline (37). Other strongly downregulated genes are the *wapA* operon, expressing one of the main cell surface proteins in *B. subtilis* (38), the *fadN* operon involved in fatty acid degradation (39), and the *frlB* operon coding for an amino sugar uptake system (40). Finally, the expression of the major citrate synthase encoded by *citZ* was also significantly downregulated (41). Several of the up- and downregulated genes in Table 1 were found in a previous transcriptome study of a Δ*whiA* mutant grown in a rich LB medium (9). A graphical comparison of both studies is depicted in Fig. S4.

In Table 1, the main genes involved in branched-chain amino acids metabolism and fatty acid synthesis are lacking, since their differential expression was lower than

threefold. The fold change difference and *P*-values of these genes are listed in Table S2. The most upregulated genes, with fold differences varying between 1.9- and 1.5-fold, were the branched-chain amino acid transporters *bcaP* and *braB* and the genes *ilvE* and *ilvD* involved in branched-chain fatty acid precursors' synthesis. Possibly, this is a response to low substrate levels. However, *yvbW*, encoding a putative leucine permease, was downregulated 1.7-fold, and the *leuA* operon involved in leucine biosynthesis was also downregulated significantly but only by approximately 1.4-fold, and there was no significant difference in expression of either valine or isoleucine biosynthesis genes (Table S2). In conclusion, the transcriptome data did not provide a clear explanation for the observed decrease in branched-chain fatty acid precursors in the Δ*whiA* mutant, and also no clues for why cell division and chromosome segregation are affected in such mutant.

## Fatty acid analysis of a Δ*whiA* mutant

Like most Gram-positive bacteria, *B. subtilis* contains primarily branched-chain fatty acids. Synthesis of anteiso-fatty acids requires 2-methylbutyrate, and the iso-C15 and -C17 and iso-C14 and -C16 fatty acids require isovalerate and isobutyrate, respectively (Fig. S5) (42). These branched-chain fatty acids are important for the regulation of membrane fluidity in Gram-positive bacteria (43). The isoform contains a methyl side chain at the second terminal C atom, whereas the anteiso form contains a methyl side chain at the third terminal C atom. Both increase the fluidity of the membrane, but the anteiso more so than the isoform. The reduced cellular concentration of branched-chain fatty acid precursors in a Δ*whiA* mutant might lead to a change in the fatty acid composition of the cell membrane. In theory, this could affect chromosome replication and cell division, since DNA replication involves the recruitment of essential components to the cell membrane, such as the DNA replication initiator protein DnaA in *E. coli* (44) and the DNA helicase loader DnaB in *B. subtilis* (45). Naturally, cell division is also closely associated with the cell membrane. In fact, the association of the *E. coli* cell division regulator MinD with the cell membrane requires unsaturated acyl chains (46), presumably because this facilitated the insertion of the C-terminal amphipathic helix of MinD, which functions as a membrane anchor. Insertion of such a relatively bulky helix in between phospholipid molecules will cost less energy when lipids are less densely packed, and the membrane is in a more fluid state (47). The *B. subtilis* FtsZ membrane anchors, SepF and FtsA, also use an amphipathic helix to bind to the cell membrane (48, 49). Changes in fatty acid composition can, therefore, influence cell division. To examine whether the fatty acid composition of Δ*whiA* cells differs from wild-type cells, we performed a gas chromatography-based analysis (Table S3). For this, cells were harvested when the cultures reached an $OD_{500}$ of approximately 0.5, when cell division and chromosome segregation defects became visible. The majority of fatty acids, 93.9% in the wild-type strain, are branched-chain fatty acids, and this fraction hardly changes in the Δ*whiA* mutant (93.1%). The absence of WhiA results in the reduction of iso-fatty acids from 53.8% to 45.8 %, whereas the fraction of anteiso-fatty acids increases from 40.1% to 47.3 % (Fig. 7). The reduction in isovalerate and isobutyrate derived fatty acids is in line with the metabolome data, but the increased contribution of the 2-methylbutyrate-derived fatty acids is not. Anteiso-fatty acids disturb the lipid packing more than iso-fatty acids and will therefore increase membrane fluidity, which is an important way *B. subtilis* regulates its membrane fluidity (50). This might explain why the Δ*whiA* mutant contains 2.5% less short fatty acid species (C13, C14, and C15) and 3.6% more long fatty acid species (C16, C17, and C18) (Fig. 7) so that membrane fluidity homeostasis is maintained. Indeed, a membrane fluidity assay, using the membrane fluidity-sensitive dye Laurdan (51, 52), did not detect significant differences in membrane fluidity between both strains when grown in either minimal or LB medium (Table S4). We also checked whether the addition of branched-chain fatty acid precursors (isovalerate, isobutyrate, and 2-methyl butyrate) to LB growth medium restored the growth defect of a Δ*whiA* mutant, but this was not the case (data not

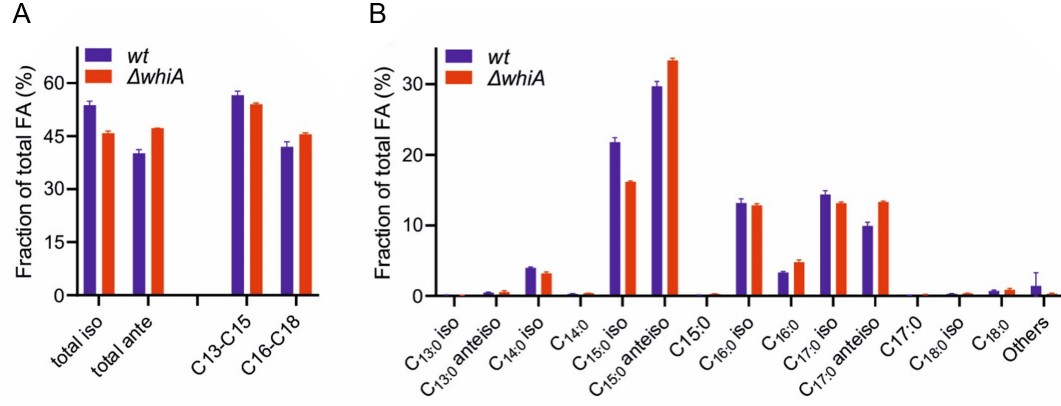

**FIG 7** Fatty acid analysis of a Δ*whiA* mutant. (A) Comparison of the total iso- and anteiso-fatty acids and fatty acid chain length between wild-type *B. subtilis* and Δ*whiA* cells. (B) Details of the different fatty acids. Concentrations of individual fatty acids are listed in Table S3.

included). It seems therefore unlikely that the phenotypes of a Δ*whiA* mutant can be explained by changes in the fatty acid composition of the cell membrane.

## Changes in chromosome conformation in a Δ*whiA* mutant

Polymerization of FtsZ over the area occupied by the nucleoid is inhibited by the protein Noc in *B. subtilis*. Noc binds to multiple sites on the chromosome but also associates with the cell membrane. It is believed that the nucleoprotein complexes formed at the membrane periphery by Noc physically interfere with efficient FtsZ polymerization (53); in this way, nucleoid occlusion links cell division to chromosome segregation. Noc is not essential, however, when both Noc and WhiA are absent, cells become very filamentous and grow extremely poorly (9). Since a Δ*whiA* mutant shows increased spacing between nucleoids, it is conceivable that the protein is somehow involved in altering the chromosome organization. One of the main nucleoid organizing complexes is the DNA clamp-forming SMC complex (54, 55). In many bacteria, the SMC complex moves along the DNA from the origin to the terminus and thereby tethers the left and the right arms of the chromosome (54). Interestingly, a previous ChIP-on-chip study with WhiA revealed a genome binding profile that showed some overlap with the genome binding profile of SMC (9, 56). To examine whether the activity of SMC is disturbed in a Δ*whiA* mutant, we performed a Hi-C (chromosome conformation capture) analysis of wild-type and Δ*whiA* cells. As shown in the contact map of Fig. 8A, it is clear that the absence of WhiA does not affect the juxtaposition of chromosome arms by the SMC condensin complex (57). However, a closer inspection of the difference in chromosome contacts indicated that short-range interactions (< 50 kb) are reduced in the absence of WhiA (Fig. 8B), possibly suggesting that WhiA has a DNA organization role and is involved in altering local chromosomal interactions.

## Conclusion

Despite the conserved nature of WhiA and its documented role as a transcriptional activator in actinomycetes, it is unclear how this protein functions in *B. subtilis* and other Gram-positive non-actinomycetes. In this study, we were able to eliminate three possible mechanisms, including an effect on central carbon metabolism, changes in the fatty acid composition of the cell membrane, and organization of the chromosome by the SMC condensin complex. Interestingly, we found that the absence of WhiA affects the compaction of the chromosome. The resolution of Hi-C is currently insufficient to correlate the differences in DNA packing with either WhiA binding sites on the chromosome or transcriptome data. It will be interesting to see whether this new phenotype of

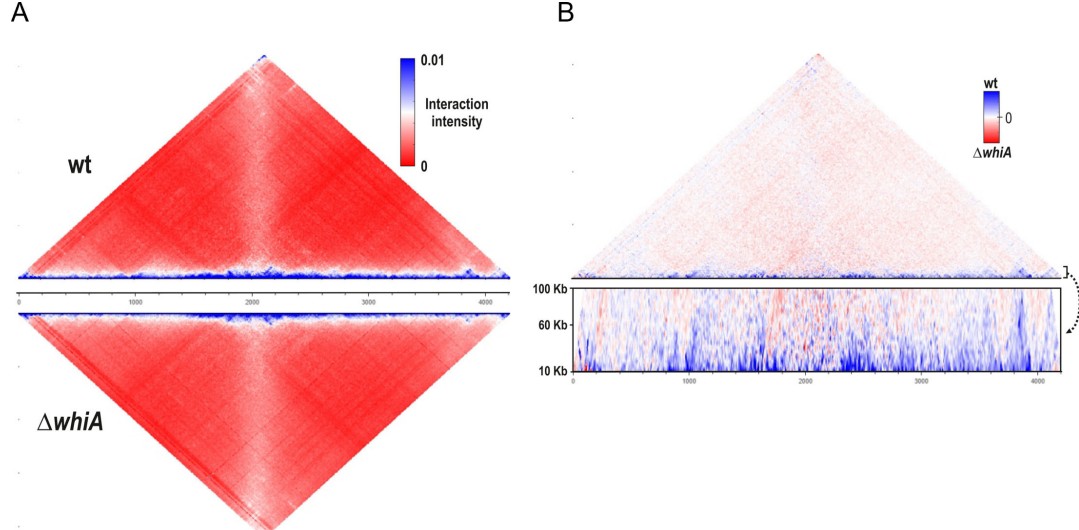

**FIG 8** Chromosome conformation capture (Hi-C) analysis. (A) Normalized Hi-C contact maps of wild-type (top) and Δ*whiA* strains (below) at the exponential phase. SMC-dependent juxtaposition of the chromosome arms is observed in both strains as the secondary (vertical) diagonal. (B) Difference plot of wild-type and Δ*whiA* strains. The magnified view of the difference in short-range contacts (between 10 and 100 kb) is shown below.

a Δ*whiA* mutant can be linked to transcription effects, and more importantly, cell division and/or chromosome segregation.

## MATERIALS AND METHODS

### Bacterial strains and growth conditions

Luria-Bertani medium was used for routine selection and maintenance of *B. subtilis* and *E. coli* strains. Spizizen minimal medium (58) consisted of 2 g/L $(NH_4)_2SO_4$, 14 g/L $K_2HPO_4$, 6 g/L $KH_2PO_4$, 1 g/L sodium citrate, 2 g/L $MgSO_4$, 5 g/L glucose, 2 g/L tryptophan, 0.2 g/L casamino acids, and 2.2 g/L ammonium ferric citrate. The defined minimal Amber medium consisted of 70 mM $K_2HPO_4$, 30 mM $KH_2PO_4$ (adjusted to pH 7.4), 15 mM sodium chloride, 10 mM $(NH_4)_2SO_4$, 0.002 mM of trace elements ($ZnCl_2$, $MnSO_4$, $CuCl_2$, $CoCl_2$, and $Na_2MoO_4$), 22 mM glucose, 0.25 mM tryptophan, 10 mM glutamate, 1 mM $MgSO_4$, 0.1 mM calcium chloride, and 0.01 mM ammonium ferric citrate (Table S1). When indicated, the medium was supplemented with 22 mM final concentration of malate, fumarate, or citrate. All strains were grown at 37°C at 250 rpm. Growth in AMBER and LB medium was followed by measuring the optical density using 500 and 600 nm, respectively. *B. subtilis* strains used in this study are listed in Table S5. When indicated, LB medium was supplemented with a mixture of three branched-chain fatty acid precursors (100 µM of 2-methyl-butyrate, isobutyrate, and isovalerate, Sigma-Aldrich) or straight fatty acid precursors (100 µM of methyl-butyrate, methyl-propionate, and methyl-valerate, Sigma-Aldrich).

WhiA depletion strain (LB45) (10) was always grown in the presence of erythromycin, due to the Campbell-type integration of the P*spac-whiA* construct into the *whiA* locus. Depletion of WhiA was accomplished by inoculating a single colony into LB medium with 0.1 mM IPTG and growth at 37°C to an $OD_{600}$ of ~1. Subsequently, cells were harvested, washed in pre-warmed LB medium, resuspended to an $OD_{600}$ of 0.01, and grown in the absence of IPTG.

## Microscopy

Exponentially growing cells were stained with the fluorescent membrane dye FM-95 (0.5 µg/mL final concentration), and the DNA was stained with DAPI (1 µg/mL final concentration), and after 5-min incubation mounted on microscope slides covered with a thin film of 1% agarose. Microscopy was performed on an inverted fluorescence Nikon Eclipse Ti microscope equipped with a CFI Plan Apochromat DM 100× oil objective, an Intensilight HG 130bW lamp, and a C11440‐22CU Hamamatsu ORCA camera. The digital images were acquired and analyzed with ImageJ v.1.48d5 (National Institutes of Health). Cell lengths were measured manually using ImageJ software.

## Metabolome analysis

The main culture (20 mL) was inoculated with an exponentially growing overnight culture to an initial $OD_{500}$ of 0.05. The optical density was monitored and 2 mL cell suspension was sampled. Three experiments were carried out to provide the necessary biological replicates. During cultivation, the pH value was determined at each sampling time point by using HI 2211 pH/mV/uC bench meter (Hanna Instruments Deutschland GmbH, Kehl, Germany). A total of 2 mL of cell culture medium was taken at 60, 120, 180, 240, 300, 360, 420, and 480 min by sterile filtration, using a 0.45 mm pore size filter (Sarstedt AG, Nuernberg, Germany), to get sterile extracellular metabolite samples of the bacterial culture and directly frozen until measurement. $^1$H-NMR analysis was carried out as described previously (59). In brief, 400 µL of the sample was mixed with 200 µL of sodium hydrogen phosphate buffer (0.2 M, pH 7.0) to avoid chemical shifts due to pH, which was made up with 50% $D_2O$. The buffer also contained 1 mM trimethylsilyl propanoic acid-$d_4$ (TSP), which was used for the quantification and also as a reference signal at 0.0 ppm. To obtain NMR spectra, a 1D-NOESY pulse sequence was used with a presaturation on the residual partly deuterated water (HDO) signal. A total of 64 FID scans were performed at 600.27 MHz and at a temperature of 310 K using a Bruker AVANCE-II 600 NMR spectrometer operated by TOPSPIN 3.1 software (both from Bruker Biospin). For qualitative and quantitative data analysis, we used AMIX (Bruker Biospin, version 3.9.14). We used the AMIX Underground Removal Tool on obtained NMR spectra to correct the baseline, thereby using the following parameters: left border region 20 ppm, right border region −20 ppm, and a filter width of 10 Hz. Absolute quantification was performed as previously described (59). In brief, a signal of the metabolite, either a complete signal or a proportion, was chosen manually and integrated. The area was further normalized on the area of the internal standard TSP and on the corresponding number of protons and the sample volume. For statistical comparison of extracellular metabolite data and growth, bar charts, and XY plots, we used Prism (version 6.01; GraphPad Software). The time-resolved extracellular metabolite concentrations were $log_2 (x + 1)$ transformed for the separation via PCA. The PCA was done using PAST v3.16 with auto-scaled data (60).

## Transcriptome analysis

Cells (2 mL cultures) were spun down (30 s Eppendorf centrifuge, 14,000 rpm, 4°C), resuspended in 0.4 mL ice-cold growth medium, and added to a screw cap Eppendorf tube containing 1.5 g glass beads (0.1 mm), 500 µL phenol:chloroform:isoamyl alcohol (25:24:1), 50 µL 10% SDS, and 50 µL RNase-free water (61). All solutions were prepared with diethylpyrocarbonate-treated water. After vortexing, tubes were frozen in liquid nitrogen and stored at −80°C. Cells were broken using a bead-beater for 4 min at room temperature. After centrifugation, the water phase was transferred to a clean tube containing 400 µL chloroform; after vortexing and centrifugation, the water phase was used for RNA isolation with High Pure RNA Isolation Kit (Roche Diagnostics GmbH, Mannheim, Germany), yielding >3 µg total RNA per sample. TapeStation System (Agilent) was used for checking the integrity of the RNA, and integrity number (RNA) values of 8.3–9.2 were obtained. For next-generation sequencing, a ribosomal RNA

depletion was performed on the total RNA using the Ribo-Zero rRNA Removal Kit (Gram-Positive Bacteria, Illumina). Bar-coded RNA libraries were generated according to the manufacturer's protocols using the Ion Total RNA-Seq Kit v2 and the Ion Xpress RNA-Seq barcoding kit (Thermo Fisher Scientific). The size distribution and yield of the barcoded libraries were assessed using the 2200 TapeStation System with Agilent D1000 ScreenTapes (Agilent Technologies). Sequencing templates were prepared on the Ion Chef System using the Ion PI Hi-Q Chef Kit (Thermo Fisher Scientific). Sequencing was performed on an Ion Proton System using an Ion PI v3 chip (Thermo Fisher Scientific) according to the instructions of the manufacturer. After quality control and trimming, the sequence reads were mapped onto the genome (genome-build-accession NCBI Assembly: GCA_000009045.1) using the Torrent Mapping Alignment Program. The Ion Proton system generates sequence reads of variable lengths, and this program combines a short read algorithm (62) and long read algorithms (63) in a multistage mapping approach. The gene expression levels were quantified using HTseq (64). The data were normalized and analyzed for differential expression using R statistical software and the DESeq2 package (65). The RNA-seq data have been submitted to and are accessible in the Gene Expression Omnibus using accession number GSE121479.

## Lipid analysis

The fatty acid composition was determined from cells growing in Amber medium when the cultures reached an $OD_{600}$ of approximately 0.5. Cells were harvested by centrifugation at $10,000 \times$ rcf for 5 min at 4°C, washed once with 0.9% ice-cold NaCl, and submitted to flash freeze in liquid $N_2$. Fatty acids were analyzed as fatty acid methyl esters using gas chromatography. All analyses were carried out in triplicates at the Laboratory Genetic Metabolic Disease, Amsterdam UMC.

## Laurdan GP spectroscopy

Membrane fluidity was measured as described before using Laurdan generalized polarization (GP) (52). Cells were grown in either LB or SMM to an OD of approximately 0.5, followed by 5-min incubation with 10 mM Laurdan. Subsequently, cells were washed three times with pre-warmed buffer containing 50 mM $Na_2HPO_4/NaH_2PO_4$ pH 7.4, 0.1% glucose, and 150 mM NaCl with and without the membrane fluidizer benzyl alcohol (30 mM). The Laurdan fluorescence intensities were measured at 435 ± 5 nm and 490 ± 5 nm upon excitation at 350 ± 10 nm, using a Tecan Infinite 200M fluorometer. The Laurdan GP was calculated using the formula $GP = (I_{435} - I_{490})/(I_{435} + I_{490})$.

## Chromosome capture by Hi-C

Cultures were grown in LB media with shaking, and samples for Hi-C were collected at the exponential growth phase ($OD_{600}$: 0.6). Hi-C was carried out exactly as described before with digestion, using HindIII (66). Hi-C matrices were constructed using the Galaxy HiCExplorer webserver (67). Briefly, paired-end reads were mapped separately to the *B. subtilis* genome (NCBI Reference Sequence: NC_000964.3) using very sensitive local setting mode in Bowtie2 (Galaxy v.2.3.4.2). The mapped files were used to build the contact matrix using the tool hicBuildMatrix (Galaxy v.2.1.2.0) using a bin size of 10 kb, and HindIII restriction site (AAGCTT) and AGCT as the dangling sequence. The contact matrix was then used for further analysis and visualization using the interactive browser-based visualization tool "Bekvaem" essentially as described before (66).

## ACKNOWLEDGMENTS

We would like to thank Henrik Strahl (Newcastle University) for scientific assistance with the lipid analysis and for insightful discussions, Johan Westerhuis (University of Amsterdam) for his insights about the analysis of the exometabolome, Selina van Leeuwen (MAD, UvA) for providing excellent sequencing services, and members of the group, and the Advanced Multidisciplinary Training in Molecular Bacteriology (AMBER)

EU Marie Curie Initial Training Network (ITN), for insightful discussions, and Gertjan Kramer and Stanley Brul for critical reading of the manuscript.

The research was funded by EU Marie Curie ITN grant AMBER (317338), Marie Curie CIG grant DIVANTI (618452), European Commission MCSA-IF 749510, and STW Vici grant 12128.

## AUTHOR AFFILIATIONS

[1]Swammerdam Institute for Life Sciences, University of Amsterdam, Amsterdam, the Netherlands
[2]Institute of Biochemistry, University of Greifswald, Greifswald, Germany
[3]Laboratoire de Genetique Microbienne, Domaine de Vilvert, Institut National de la Recherche Agronomique, Jouy-en-Josas, France
[4]RNA Biology and Applied Bioinformatics Research Group, Swammerdam Institute for Life Sciences, University of Amsterdam, Amsterdam, the Netherlands

## AUTHOR ORCIDs

Leendert W. Hamoen http://orcid.org/0000-0001-9251-1403

## FUNDING

| Funder | Grant(s) | Author(s) |
| --- | --- | --- |
| EU Marie Curie ITN AMBER | 317338 | Marie-Françoise Noirot-Gros |
| | | Michael Lalk |
| | | Leendert W. Hamoen |
| EU Marie Curie CIG DIVANTI | 618452 | Leendert W. Hamoen |
| EU Marie Curie IF | 749510 | Gaurav Dugar |
| NWO STW Vici | 12128 | Leendert W. Hamoen |

## AUTHOR CONTRIBUTIONS

Laura C. Bohorquez, Data curation, Formal analysis, Investigation, Validation, Visualization, Writing – original draft | Joana de Sousa, Formal analysis, Investigation, Visualization, Writing – original draft | Transito Garcia-Garcia, Formal analysis, Investigation | Gaurav Dugar, Formal analysis, Investigation, Visualization, Writing – original draft | Biwen Wang, Formal analysis, Investigation, Visualization, Writing – original draft | Martijs J. Jonker, Data curation, Formal analysis, Writing – original draft | Marie-Françoise Noirot-Gros, Formal analysis, Investigation, Supervision, Writing – original draft, Writing – review and editing | Michael Lalk, Data curation, Formal analysis, Investigation, Supervision, Writing – original draft, Writing – review and editing, Resources | Leendert W. Hamoen, Conceptualization, Funding acquisition, Supervision, Writing – review and editing

## ADDITIONAL FILES

The following material is available online.

### Supplemental Material

**Supplemental material (Spectrum01795-23-s0001.pdf).** Tables S1 to S5 and Fig. S1 to S5.
**Table S6 (Spectrum01795-23-s0002.xlsx).** Transcriptome differences between whiA mutant and wild-type cells.

Open Peer Review

**PEER REVIEW HISTORY (review-history.pdf).** An accounting of the reviewer comments and feedback.

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
