## [Reviewer comments · Microbiology Spectrum]

Microbiology Spectrum

Metabolic and chromosomal changes in a *Bacillus subtilis* *whiA* mutant

Laura Bohorquez, Joana de Sousa, Transito Garcia-Garcia, Gaurav Dugar, Biwen Wang, Martijs Jonker, Marie-Francoise Noirot-Gros, Michael Lalk, and Leendert Hamoen

Corresponding Author(s): Leendert Hamoen, Universiteit van Amsterdam

Review Timeline:

Submission Date:	April 28, 2023
Editorial Decision:	August 10, 2023
Revision Received:	September 20, 2023
Accepted:	October 10, 2023

Editor: Jing Han

Reviewer(s): The reviewers have opted to remain anonymous.

Transaction Report:

DOI: <https://doi.org/10.1128/spectrum.01795-23>

August 10, 2023

Prof. Leendert W. Hamoen
Universiteit van Amsterdam
Bacterial Cell Biology, Swammerdam Institute for Life Sciences (SILS)
Science Park 904
Amsterdam 1098 XH
Netherlands

Re: Spectrum01795-23 (Metabolic and chromosomal changes in a *Bacillus subtilis* whiA mutant)

Dear Prof. Leendert W. Hamoen:

Link Not Available

Sincerely,

Jing Han

Journals Department
Reviewer comments:

Reviewer #1 (Comments for the Author):

In general the experiments appear to have been carefully conducted and the text is mostly comprehensibly written, but needs further editing in language and style. My main concern however, is the fact that the experimental data is very rich, but there are few positive results. The results showed that WhiA neither functions by regulating central carbon metabolism nor affects fatty acid composition. Only chromosome conformation capture (Hi-C) analysis indicated short range interactions are reduced in the absence of WhiA. In order to reveal the mechanism of WhiA in *B. subtilis*, it is suggested that the authors should conduct in-depth research from the perspective of compaction of the chromosome.

Reviewer #2 (Comments for the Author):

The study by Bohorquez et al uses whole cell investigative experiments to understand the function of *whiA* in *Bacillus subtilis*. They identify a cell division and DNA segregation defect in *zapA/whiA* mutant in minimal media, changes in BCFA precursor levels using metabolomics, changes in transcriptomics that did not correlate with metabolome data, changes in fatty acid composition which does not translate to changes in membrane fluidity, and a reduction in short range chromosome interactions. This study highlights the complexity of whole cell studies to understand the function of single gene, as well as the interconnectedness of metabolism, DNA replication and segregation, and cell division. The study is well presented and logically explained, but requires a few modifications.

Feedback:

Figure 2: coloured microscopy images do not show fluorescent membrane labelling, they show phase contrast image pseudo-coloured red, with DAPI in cyan. Images need to be corrected, or description updated.

Knockdown of *whiA* in *zapA* deletion strain - did you confirm changes in expression? (Western blot or qPCR) Similarly, was there any polar effects on *glmR* and *crh* when *whiA* was knocked down?

In the *zapA* mutant, chromosomal defects and minicells observed when *whiA* depleted - does FtsZ localise over chromosomes in this double mutant? Are Z rings misplaced?

The membrane fluidity assay described is in methods and discussed in results, but no data is included. This could be included in Figure 7.

The data for the growth rescue of *whiA* mutant when BCFA are added is also not shown - this could be a supplementary figure, or stated that data is not included. Was this growth rescue performed in LB?

Methods - details lacking for microscopy section that needs to be expanded. There is no reference for concentrations and incubation times for fluorescent probe labelling. Do you mean FM-595 was used to label membranes? Information about transmitted light method is missing - presumably phase contrast was used? Information about objective magnification and numerical aperture is missing. No details on image analysis are included - how were cell lengths and internucleoid distances measured? Manually? What parameters were used?

Minor comments:

Line 25 - spelling: motif

Figure 1 - graphs show both OD600 and OD500 - is this correct?

Lines 110, 115 and 116 reference Figure 1A but no alphabetical labelling is included in Figure 1

Lines 308 and 309 - Do you mean μM for FA concentrations?

Staff Comments:

Preparing Revision Guidelines

Please return the manuscript within 60 days; if you cannot complete the modification within this time period, please contact me. If you do not wish to modify the manuscript and prefer to submit it to another journal, please notify me of your decision immediately so that the manuscript may be formally withdrawn from consideration by Microbiology Spectrum.

Corresponding authors may join or renew ASM membership to obtain discounts on publication fees. Need to upgrade your

membership level? Please contact Customer Service at Service@asmusa.org.

Point-by-point response to reviewers comments

We would like to begin by thanking the reviewers for their efforts and useful comments. For clarity, our replies are presented in **blue**.

Reviewer #1 (Comments for the Author):

In general the experiments appear to have been carefully conducted and the text is mostly comprehensibly written, but needs further editing in language and style. My main concern however, is the fact that the experimental data is very rich, but there are few positive results. The results showed that WhiA neither functions by regulating central carbon metabolism nor affects fatty acid composition. Only chromosome conformation capture (Hi-C) analysis indicated short range interactions are reduced in the absence of WhiA. In order to reveal the mechanism of WhiA in *B. subtilis*, it is suggested that the authors should conduct in-depth research from the perspective of compaction of the chromosome.

We agree with the reviewer that part of our results describe the disproval of two working models for the conserved regulator WhiA (a role in central carbon metabolism and regulation of fatty acid composition). However, this information is important since these were plausible working models, and therefore helps to focus the effort to elucidate the working mechanism of WhiA. In that sense, the disproval of these two working models can be considered positive results. In addition, we do provide a lead for further research and that is a role of WhiA in the organization of the chromosome. Of course, this now needs to be further investigated in detail. Unfortunately, the tools to study chromosome organization at the next level are not trivial, and will require a whole new PhD/postdoc project, that goes beyond the current project. We hope we can perform such in-depth study in the future.

Reviewer #2 (Comments for the Author):

The study by Bohorquez et al uses whole cell investigative experiments to understand the function of whiA in *Bacillus subtilis*. They identify a cell division and DNA segregation defect in zapA/whiA mutant in minimal media, changes in BCFA precursor levels using metabolomics, changes in transcriptomics that did not correlate with metabolome data, changes in fatty acid composition which does not translate to changes in membrane fluidity, and a reduction in short range chromosome interactions. This study highlights the complexity of whole cell studies to understand the function of single gene, as well as the interconnectedness of metabolism, DNA replication and segregation, and cell division. The study is well presented and logically explained, but requires a few modifications.

Feedback:

Figure 2: colorised microscopy images do not show fluorescent membrane labelling, they show phase contrast image pseudo-coloured red, with DAPI in cyan. Images need to be corrected, or description updated.

We would like to thank the reviewer for pointing out this mistake. Indeed the red background

in the images is not membrane stain but pseudocolored phase contrast images to increase the contrast with the DAPI stained nucleoids. We have now indicated this clearly in the legend of Fig. 2 (lines 618-625 in the revision).

Knockdown of *whiA* in *zapA* deletion strain - did you confirm changes in expression? (Western blot or qPCR) Similarly, was there any polar effects on *glmR* and *crh* when *whiA* was knocked down?

The *WhiA* depletion strain has been extensively documented in previous publications (Surdova et al. 2013 PMID: 24097947, Bohorquez et al. 2028 PMID: 29378890), and we have not checked again this depletion. From the previous work we know that it takes some time before morphological changes, such as cell elongation, become visible, as we also have observed in this study and show in Fig. 2A and 2B, and have mentioned in the text. In addition, in the Surdova et al. 2013 PMID: 24097947 paper we have extensively checked that the cell division phenotype was not related to any polar effect on *crh*. We have added this information now to the main text (line 128 in the revision). With respect to *glmR*, this gene is located upstream of *whiA* and therefore not affected by the *Pspac* insertion used in the *WhiA* depletion strain (Surdova et al. 2013 PMID: 24097947). This misunderstanding is possibly a consequence of the fact that we have not clearly explained the organization of the operon. We do this now in line 101 of the revision.

In the *zapA* mutant, chromosomal defects and minicells observed when *whiA* depleted - does FtsZ localise over chromosomes in this double mutant? Are Z rings misplaced?

Upon depletion of *WhiA* in a *zapA* mutant cells become elongated and Z rings disappear. We have shown this in Surdova et al. 2013 PMID: 24097947. But, we agree that it would be good to mention this clearly in the text and we have added this information in line 120 of the revision.

The membrane fluidity assay described is in methods and discussed in results, but no data is included. This could be included in Figure 7.

We present the Laurdan generalized polarization results, used to measure membrane fluidity, now in the new Table S4 (referred to in line 257 of the revision). Since the Laurdan GP values did not differ between wild type cells and the *whiA* mutant, we feel this data is better placed in the SI.

The data for the growth rescue of *whiA* mutant when BCFA are added is also not shown - this could be a supplementary figure, or stated that data is not included. Was this growth rescue performed in LB?

We mention now in the main text "data no included" (line 259 in the revision), and indicate that the BCFA precursor were added to the LB medium (line 258 in the revision).

Methods - details lacking for microscopy section that needs to be expanded. There is no reference for concentrations and incubation times for fluorescent probe labelling. Do you mean FM-595 was used to label membranes? Information about transmitted light method is

missing - presumably phase contrast was used? Information about objective magnification and numerical aperture is missing. No details on image analysis are included - how were cell lengths and internucleoid distances measured? Manually? What parameters were used?

We have now provided more details on the microscopy techniques used, including, the concentration of the fluorescent membrane and DNA dyes, incubation time and details on the microscopic setup (lines 320-327 of the revised manuscript).

Line 25 - spelling: motif

Thank you for pointing this out. The spelling mistake has been corrected.

Figure 1 - graphs show both OD600 and OD500 - is this correct?

Thank you for pointing this out. It should have been OD₅₀₀ for both graphs. We have now corrected this, and also explained in the Material and Methods when we use OD500 and OD600 (line 308 in the revision).

Lines 110, 115 and 116 reference Figure 1A but no alphabetical labelling is included in Figure 1

Thank you for pointing out this typo. No alphabetical labelling is necessary in Fig. 1. We have now removed this from the text (lines 110, 115 and 116 in the revision).

Lines 308 and 309 - Do you mean μM for FA concentrations?

Thank you for pointing out this typo. Yes indeed it should have been μM . We have now corrected this (lines 310 and 312 in the revision).

October 10, 2023

Prof. Leendert W. Hamoen
Universiteit van Amsterdam
Bacterial Cell Biology, Swammerdam Institute for Life Sciences (SILS)
Science Park 904
Amsterdam 1098 XH
Netherlands

Re: Spectrum01795-23R1 (Metabolic and chromosomal changes in a *Bacillus subtilis* whiA mutant)

Dear Prof. Leendert W. Hamoen:

Your manuscript has been accepted, and I am forwarding it to the ASM Journals Department for publication. You will be notified when your proofs are ready to be viewed.

Sincerely,

Jing Han
Editor, Microbiology Spectrum
